# Defining Attachment and Bonding: Overlaps, Differences and Implications for Music Therapy Clinical Practice and Research in the Neonatal Intensive Care Unit (NICU)

**DOI:** 10.3390/ijerph18041733

**Published:** 2021-02-10

**Authors:** Mark Ettenberger, Łucja Bieleninik, Shulamit Epstein, Cochavit Elefant

**Affiliations:** 1Department of Music Therapy, University Hospital Fundación Santa Fe de Bogotá, Bogotá 110111, Colombia; 2SONO—Centro de Musicoterapia, Bogotá 110221, Colombia; 3Institute of Psychology, University of Gdansk, 80-309 Gdansk, Poland; lucja.bieleninik@ug.edu.pl; 4GAMUT—The Grieg Academy Music Therapy Research Centre, NORCE Norwegian Research Centre AS, 5029 Bergen, Norway; 5School for Creative Arts Therapies, University of Haifa, Haifa 3498838, Israel; shushuq@gmail.com (S.E.); celefant@univ.haifa.ac.il (C.E.)

**Keywords:** music thAerapy, bonding, attachment, preterm infants, Neonatal Intensive Care Unit (NICU), family-centered care

## Abstract

Preterm birth and the subsequent hospitalization in the Neonatal Intensive Care Unit (NICU) is a challenging life event for parents and babies. Stress, anxiety, and depressive symptoms, limitations in holding or touching the baby, and medical complications during the NICU stay can negatively affect parental mental health. This can threaten the developing parent-infant relationship and might adversely impact child development. Music therapy in the NICU is an internationally growing field of clinical practice and research and is increasingly applied to promote relationship building between parents and babies. The two most commonly used concepts describing the early parent-infant relationship are ‘attachment’ and ‘bonding’. While frequently used interchangeably in the literature, they are actually not the same and describe distinctive processes of the early relationship formation. Thus, it is important to discuss the overlaps and differences between attachment and bonding and the implications for music therapy clinical practice and research. Whereas providing examples and possible scenarios for music therapists working on either bonding or attachment, the distinction between both concepts is relevant for many health care professionals concerned with early parenting interventions in the NICU. This will hopefully lead to a more precise use of theory, and ultimately, to a more informed clinical practice and research.

## 1. Introduction

The formation of positive emotional bonds between parents and their baby is one of the most significant foundations in the construction of a healthy parent–infant relationship and fundamental for the newborn’s development later on in life [1,2]. This is a process that begins early in pregnancy, lasting throughout its entire period, and continues after childbirth [3,4]. How both mothers and fathers feel towards their unborn baby and the behaviors they show is influenced by many factors, including parental mental health, social support, complications during pregnancy, or previous pregnancies, among others [5]. While individual differences exist in terms of timing, duration, and form, the maternal–fetal relationship usually intensifies over time, although many mothers experience a significant shift when they notice the first fetal movements in the transition from the 2nd to the 3rd trimester [6,7]. This is also the time when the baby progressively reacts to external stimuli such as the mother’s voice and touch [8,9] and thus it is no surprise that parents increasingly start interacting with their baby during this time. The paternal–fetal relationship is not yet as well investigated, but it has been shown that also men go through physiological and psychological changes in their journey towards fatherhood, which influence the quality of their interactions with the newborn after birth [10]. Additionally, the emotional bonds that parents form with the fetus are also affected by the early relational experiences with their own primary caregivers. Thus, the parents’ own attachment representations can have an effect on their behavior pre- and postnatally and constitute some of the earliest moderators for child development [6,10].

When a baby is born preterm (defined as less than 37 weeks of gestation), this process can get interrupted. Often, preterm birth appears suddenly and unexpectedly, in a time when parents might not yet be psychologically prepared to become parents [11]. Preterm delivery differs a great deal from how parents imagine the labor situation and regularly takes place in an emergency context. Lack of control, uncertainty about the baby’s health, medical complications in the course of the delivery, and a possible hospitalization of the mother herself may furthermore add emotional burden to parents postpartum. In the Neonatal Intensive Care Unit (NICU), having a preterm baby needing life sustaining medical treatment can trigger depressive symptoms in parents [12] or increase the risk for developing postpartum depression [13]. Anxiety, guilt, shame, anger, resignation, and the disruption of family life are further challenges that may alter the transition to parenthood or can cause symptoms of Post-traumatic Stress Disorder or Acute Stress Disorder [14,15,16]. This situation can pose a threat to the continuum or stability of the emerging parent–infant relationship. Consequently, several studies and reviews have highlighted the risks for an impaired bonding in parents of preterm babies and the possible repercussions for child development [17,18,19]. Yet, there is also contrasting information on this topic and not all parents experience these challenges similarly. A recent meta-analysis shows for example just slightly elevated stress levels in parents of preterm babies [20] and other studies found no differences in maternal attachment representations or in maternal sensitivity when compared to parents of full-term babies [21,22]. Thus, relationship building in the NICU requires a differential view. While the hospitalization in the NICU might be a stressor that can trigger parental mental health challenges and negatively affect the parent–infant relationship, coping mechanisms differ from family to family, and even from mothers to fathers [23]. Thus, an individualized view is required, taking into account the parents’ own relational experiences, occurrences during pregnancy, their coping mechanisms, resilience, social support networks, mental health history, and socio-demographic factors.

This article will discuss two of the most commonly used concepts when describing the early parent-infant relationship: attachment and bonding. The overlaps and differences between both concepts are highly relevant for health care professionals, including music therapists. Distinctions in focus, measurements and definitions will be outlined and examples of possible scenarios for music therapists working on either attachment or bonding will be described.

## 2. Early Parenting Interventions and Family-Centered Music Therapy in the NICU

In recent decades, there has been increasing acknowledgement of the importance of early parenting interventions in the NICU. Such interventions usually focus on improving parental mental health, infant development, and on promoting healthy parent–infant interactions [24]. Music therapy (MT) has been applied in the NICU for several decades, but the integration of family-centered care models is a newer phenomenon [25]. In coherence with other early parenting interventions, family-centered MT addresses both infants’ and parents’ needs, as well as the emerging relationship between them [26]. While recent research confirms the beneficial effects of MT on immediate or short-term outcomes for infants during their NICU stay [27,28,29,30], assessing the parent-infant relationship is one of the main research gaps in the literature [31]. To date, only a handful of studies have evaluated the impact of MT in this regard and these studies report mixed results. Parent–infant bonding improved but did not differ significantly for parents who received MT in two small-scale Randomized Controlled Trials (RCTs) [32,33]. In two other studies, parents improved their bonding scores from pre- to post-intervention timepoints, but statistical significance was not achieved [34,35]. A non-randomized controlled study, which assessed parent–infant interactions at one-month post-treatment, found no significant differences between MT and control groups [36]. 

Thus, a systematic understanding of how MT contributes to the relationship formation between parents and preterm babies is still lacking. However, a variety of models, theories and interventions exist within the field of neonatal music therapy and specific focus on bonding and attachment can be found for example in the Rhythm, Breath, Lullaby (RBL) model [37], in Creative Music Therapy (CMT) [38], or in the before-mentioned family-centered music therapy approaches [26]. With respect to research, a current international multi-center randomized controlled trial entitled is also beginning to address this gap in knowledge by assessing parental bonding during hospitalization and up to 12 months of corrected age [39].

## 3. Attachment versus Bonding

Taking this into consideration, a critical aspect to better understand how preterm birth affects the parent–infant relationship, and if MT might act as a supportive factor in this process, is to clearly determine the phenomena under discussion.

Relationship building is a multifactorial and complex process and two of the most commonly used concepts in this regard are ‘attachment’ and ‘bonding’. While both concepts are often used interchangeably in the literature, they are not the same [40,41]. Since concepts are the building blocks of theories and describe specific aspects of reality, it is therefore important to first clarify the overlaps and differences between attachment and bonding.

### 3.1. Attachment

The two pioneers in describing the bases of early relationship building were John Bowlby (1907–1990) and Mary Ainsworth (1913–1999). Both contextualized the development of relationships in the social context of the family, and, in particular, concerning the mother–infant relationship. As a result, they coined the notion of ‘attachment’ as one of the fundamental processes of how relationships emerge, build up, and mature during early childhood. Initially, attachment theory started to take shape in the 1930s and 1940s as a result of research about the personality development of children who required prolonged institutional care and therefore experienced regular changes in their primary caregivers during early infancy [42]. The full theory however was not published until the late 1960s with the first volume of the book trilogy “Attachment and Loss” in 1969 [43], and the second and third volume in 1973 (“Separation Anxiety and Anger”) [44] and 1980 (“Loss, Sadness and Depression”) [45].

According to Bowlby, the survival value of attachment was not limited to physical needs. He described attachment as a spatial concept, related to the proximity of a new-born and his primary caregiver (i.e., ‘mother-figure’) during the first months of his life. ‘Attached to’ means “… that he [a child or older person] is strongly disposed to seek proximity to and contact with that individual and to do so especially in certain specified conditions” (p. 31) [42]. Additionally, the types of relational experiences that a baby encounters with its primary caregiver will shape its attachment behavior later on, described as a first a set of behaviors that the infant shows in times of stress, pain, fatigue or in the absence of the mother-figure. Once acquired, attachment behaviors can then fluctuate and reappear within different relationships and people, but regularly remain active during the complete life span.

Kraemer (p. 493) [46] describes four essential components of the attachment theory:

First, attachment is an instinctive behavior, a form of imprinting. The baby is already born with a set of behaviors towards his mother-figure, which do not need to be learned or reinforced by her.

Second, attachment is a ‘goal corrected system’, which means that attachment serves the purpose of maintaining a perception of security within the infant, in a situation in which the infant actually is safe.

Third, attachment requires the development of internal representations or working models. The infant uses these working models of the mother-figure and the surrounding world in order to foresee their behavior and to regulate its own behavior.

And fourth, ‘the secure base’. This concept refers to the fact that if maintaining security would be the only goal for the infant and if the mother-figure would be the source of this security, there would be no motivation to disrupt this contact. However, the presence of the mother-figure makes the exploratory behaviors of infants possible in the first place. Thus, external conditions always work in relation with internal, psychological conditions that seek for homeostasis in order to work properly.

Mary Ainsworth refined Bowlby’s theoretical foundations and identified initially three (and later four) different attachment styles (i.e., secure, insecure avoidant, insecure ambivalent-resistant, insecure disorganized) that can be observed during an experimental setting called the ‘Strange Situation Test’ [47]. In this test, which usually takes place when the child is about one year old, the infant is separated from his or her caregiver for a short period of time and left alone with a stranger. The response to the reunion with the parent —rather than the level of distress during the separation — is then analyzed and classified according to the four attachment styles mentioned above. Since the development of the attachment theory, an extensive body of literature demonstrates the correlation of insecure attachment styles with increased psycho-social and mental health impairments over the child’s lifespan as compared to secure attachment styles [48,49].

### 3.2. Bonding

Mother–infant bonding was first described in the 1970s by the American pediatricians Marshall H. Klaus (1927–2017) and John H. Kennell (1922–2013). From their clinical experience, Klaus and Kennell noted that some preterm babies were re-hospitalized after discharge because they were not able to thrive despite the lack of organic diseases or due to injuries caused by their parents [50]. At that time, animal studies suggested disturbances of parenting behaviors when the animal mother was separated from her offspring shortly after birth or when typical pre- or post-birth behaviors were suppressed [51]. Thus, Klaus and Kennell were especially interested in how early or prolonged separation would affect the mothers’ behaviors of preterm or hospitalized infants, and if changes in hospital practices would improve the mother–infant relationship. In their landmark study from 1972, they found that mothers who were given additional and prolonged physical contact with their newborns shortly after birth, showed more protective and interactive behaviors and had higher maternal competences compared to control mothers [52]. As a result, they put forward the hypothesis of a ‘maternal sensitive period’, which was believed to be important for the emergence and development of a proper relationship between a mother and her child [51,53]. This ‘sensitive period’ encompasses the first hours and days after delivery, in which mothers are supposedly most likely to establish strong emotional ties with their newborns. Contrarily, if this bonding process did not happen adequately or was interrupted, alterations of the baby’s development or negative maternal feelings and behaviors towards the baby were thought to be possible [54]. Formally, bonding can be defined as: “…a maternal-driven process that occurs primarily throughout the first year of an infant’s life, but may continue throughout a child’s life. It is an affective state of the mother; maternal feelings and emotions towards the infant are the primary indicator of maternal–infant bonding” [40] (p. 1319).

Since the publication of their influential book “Maternal-infant Bonding” [53], the bonding theory was quickly absorbed by both the academic and medical and popular discourse and has been a controversial topic since then. On the one hand, it led to positive changes in care philosophies and improved parental access to their newborns, including more humanized birthing practices and early physical contact. On the other hand, the idealization of a ‘perfect birth’ led to feelings of guilt and shame for the many mothers who were not able to hold their baby after birth, either due to inflexible hospital policies or due to medical complications of the mother or the baby [54]. In addition, the scientific validity of Klaus–Kennell’s concept of emotional bonding has been questioned by a number of researchers [55] and up to date, there seems to be no definite answer to whether a ‘sensitive period’ exists or not. With respect to the effectiveness of prolonged physical contact after birth (i.e., skin-to-skin contact), however, recent meta-analyses demonstrate its positive effects on maternal health and feeding, such as a decrease in the third stage of labor or a greater success and prolonged duration in first breast-feeding attempts of the baby [56,57].

### 3.3. Overlaps and Differences

As stated in the beginning of the previous section, ‘bonding’ and ‘attachment’ are frequently used interchangeably although they differ substantially from each other. Clearly, both theories describe different aspects of the same phenomenon: how relationships are formed and how early experiences with the primary caregivers influences child development. There are however four major differences that are relevant to consider:

Focus: Although both Bowlby–Ainsworth and Klaus–Kennell clearly acknowledged the reciprocity and mutual interconnectedness of the parent-infant relationship in their theories, the most important difference is on whom the focus is laid. Attachment theory describes essentially how the child builds up a relationship with its primary caregiver and bonding theory describes the feelings, thoughts, and behaviors of the parent towards the baby. Thus, they focus on different sides of the early parent–infant relationship.

Time frame: Attachment usually implies a much broader time frame and develops over the course of the first year of life. Bonding theory is based upon a specific period of time shortly after birth including hours, days, or weeks.

Proximity: While proximity is a common feature in both theories, attachment theory focuses more on proximity as a spatial concept and is related to the sensitivity and quality of the caregiver’s response to the proximity-seeking child. In bonding theory, proximity is understood physically (i.e., skin-to-skin contact) and serves primarily to enhance the parent’s acceptance of the baby after birth.

Measurements: The type of attachment style a child forms towards its primary caregiver is assessed through the Strange Situation Test when the infant is about one year old. Bonding is assessed mainly through self-rated questionnaires that parents fill out, e.g., the Mother-to-Infant Bonding Scale (MIBS), the Post-partum Bonding Questionnaire (PBQ) or the Parental Bonding Instrument (for an overview, see Perrelli [58]).

In their concept analysis, Kinsey and Hupecey [40] examine further the similarities and differences between ‘attachment’ and ‘bonding’ according to the principles of epistemology (clarity of definition); pragmatics (applicability of the concept); linguistics (consistency in use and meaning); and logic (differentiation of the concept from related concepts). They conclude that with regard to the differentiation of the two concepts… “inconsistencies in the research literature are numerous and require that clarification be made in order for concept advancement to occur. Advancement of the concept will allow researchers to utilise appropriate measurement of the concept allowing nursing interventions to be developed that will improve bonding, thus improving maternal and child outcomes.” (p. 1315).

## 4. Discussion

As we have tried to highlight, bonding and attachment share many common features, but describe essentially distinctive processes. While this confusion seems to be superficial at first, it actually might have an important impact on how early parenting interventions inclusive of MT in the NICU are designed, implemented, and evaluated. As the impetus for defining the underlying neurological mechanisms of music and vocal interventions in the NICU continue to provide evidence for their effectiveness [30,59,60], a more thorough discussion with regard to the basic aspects of the early parent-infant relationship is still lacking. While authors within the fields of developmental psychology or child psychiatry have made a call for a better distinction between bonding and attachment for many years [61], this is now being recognized in recent systematic reviews on early parenting interventions [62]. How music therapists could adjust their interventions according to one or the other concept will be exemplified below. We will first address the potential consequences for clinical practice, and later discuss the implications for research in this context.

### 4.1. Clinical Practice–NICU MT for Attachment

Attachment theory describes how the child builds up its relationship with the parents. This is a process that typically develops during the first year of life and is mutually influenced by both the infant and the parent. Therefore, taking into consideration attachment, the music therapist in the NICU might focus on parent–infant interaction and communication. This is important, because attachment is a feedback model: parental sensitivity is for example one of the most important moderator for developing a secure attachment style, but also paramount for successful communication with the baby. The baby, however, needs also to respond adequately to a fulfilled or unfulfilled need and in this way communicates to the parent that he or she has done well. Both aspects might be altered in the NICU: the first because of psychological distress; research shows for example that parents with increased anxiety levels are less sensitive to their infants’ communicational cues [63]. The latter due to an underdeveloped nervous system and less capacity for self-regulation, which makes it harder for preterm babies to clearly manifest their needs or habituate less competently to novel stimuli than infants born at term [64]. Thus, MT can be an opportunity to practice successful interaction by helping parents to be sensitive towards their infants’ states and needs and by helping infants in their self-regulation and arousal. For the first scenario, the ideal situation would obviously be working with the baby and the parent(s). Making music and singing for the baby is an excellent way to foster parental sensitivity through a process of joint observation → musical interaction → observation. At the beginning, the music therapist might observe the infant together with the parents and may ask questions to increase their involvement, such as: “How is the baby doing today?”; “What do you think he or she needs at the moment?” This can help the parent choose music more consciously, choosing music coherent with the baby’s current state and needs, for example, singing a lullaby for sleeping, or a more energetic song for helping the baby to stay alert during feeding. While singing, the music therapist can encourage the parent to observe the baby’s reactions and return an empathic response through musical or physical gestures. It is common for music therapists in this type of work to notice the infants’ subtle responses and notify the parents about it, for example by stating “Look, he was smiling”; “Have you noticed how she turned her head towards you while you were singing?”. Such observations create an opportunity for parents to learn more about their infant’s communicational abilities, as well as their ability to provide (musically and vocally) sensitive responses through adjusting the tone of their voices, the volume, or the tempo of the music, or by making pauses, among others.

Working solely with an infant, provides an opportunity for the music therapist to work on the infant’s self-regulation capacities. One important feature of such sessions is for example the idea to entrain the music or singing to the with infant’s breathing rhythm, gestures and facial expressions [29]. This supports the infant’s ability to regulate its arousal, calm down, and orient towards the music therapist. When the infant is able to express its needs with more clarity, it might be easier for parents to understand them and provide an empathic and adequate response.

### 4.2. Clinical Practice–NICU MT for Bonding

Bonding describes the affective, cognitive, and behavioral manifestations of the parents towards the baby. Thus, when music therapists take into consideration the importance on bonding, they essentially work with parents’ feelings, thoughts, or actions, and the ideal scenario might be working with the parents alone, either in individual or group settings. Initially, exploring how parents cope with the hospitalization and supporting their emotional release and processing of the early traumatic birth, can help to construct the therapeutic relationship and confidence to explore the feelings related to the baby itself. This is important, because potential negative feelings towards the baby are difficult to express for parents and are socially not well accepted. To be able to admit for example: “I feel angry with my baby”, “I regret having this baby” or “I feel like hurting my baby” (examples from the Postpartum Bonding Questionnaire [65]), are certainly affirmations that can be challenging to acknowledge. However, even more subtle issues like feeling distant to the baby or wishing to have the ‘old days’ back without the baby can cause emotional tension in mothers and fathers. Helping parents to become aware of such feelings and thoughts, for example through music guided relaxations or other receptive MT techniques, may open a door to further verbal or musical processing [66]. Validating that many parents go through contrasting emotions during the NICU stay may additionally help release pressure. Improvisational approaches, shared music making, or MT songwriting are other potential approaches in creatively, interactively, and non-judgmentally being able to express conflictive feelings or thoughts. If required, a referral to the mental health care team or an interdisciplinary approach between music therapists, psychologists and parents might be used.

When working together with the parents and the baby in the NICU, the situation is somewhat different, since parents might feel inhibited to express such emotions in front of the baby or other staff. Letting parents choose for example songs that describe how they currently feel, and if necessary, adapt parts of the lyrics, is an approach that could help parents in expressing their hopes, fears, or worries in a way that can be more appropriate for them.

### 4.3. MT Research in the NICU–Bonding or Attachment?

As outlined above, clinical practice might differ whether music therapists want to focus on bonding or attachment. However, this is also important for current and future research, in which both concepts are still blurred. An example hereof is the recently published research protocol by Yakobson et al. [67], measuring “parent-infant attachment quality” (p. 1) or parental “attachment levels” using the Maternal Postnatal Attachment Scale (MPAS). However, they state: “The MPAS is a self-report questionnaire including 19 items investigating parents’ behaviors, attitudes, and feelings toward their infant” (p. 15), which actually refers to measuring bonding and not attachment. Another example is the current protocol for a Cochrane Review about the efficacy of musical and vocal interventions to improve neurodevelopmental outcomes for preterm infants [68]. In the objectives, the authors name bonding as one of the phenomena to be measured (p. 3). However, when specifying the outcomes the they claim to measure “attachment (measured with standardized scales, e.g., Postpartum Bonding Questionnaire)“ (p. 4).

Thus, the lack of differentiating between the two concepts is striking even in high-impact studies and reviews. This could be a confounding factor and hinder the comparability of results from previous and ongoing studies. While attachment can be assessed through the Strange Situation Test, for example in a longitudinal study, during the NICU stay, bonding seems to be the more relevant concept for assessing the parent–infant relationship within a research project.

## 5. Conclusions

Bonding and attachment share many common features, but also important differences. This is so far not acknowledged in neonatal care but may have real-world implications. While the examples in this article are drawn from the field of music therapy, a better distinction could be relevant also for other health care professionals in the NICU concerned with the early parent–infant relationship, e.g., social workers, nurses, psychologists, neonatologists, among others. Undoubtedly, relationship building is a complex process and develops reciprocally between parents and their baby along a continuum starting long before and going far beyond the NICU stay. However, the particularity of having a baby hospitalized warrants special attention. Further research distinguishing better between the two concepts is needed to understand how MT and other early parenting interventions might impact the parent–infant relationship in this setting. This will hopefully lead to a more precise use of theory and terminology, and ultimately a more informed clinical practice and research.

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
