# Peer review of "Defining Attachment and Bonding: Overlaps, Differences and Implications for Music Therapy Clinical Practice and Research in the Neonatal Intensive Care Unit (NICU)"

_ijerph, 2021, doi:10.3390/ijerph18041733_

Round 1

Reviewer 1 Report

CONTENT

Nice contribution to an important topic. Your charge for adherence to greater accuracy of terminology usage is important, but is undermined by your oversight of work where this distinction occurs, such as First Sounds: Rhythm, Breath, Lullaby (Loewy et al, 2013 [ref # 58 on your ref lest]; Loewy, 2017) in its addressing of bonding and attachment in its neonate and parent/caregiver prongs respectively. One sentence of acknowledgement would strengthen the charge for more adherence in the future. 

GRAMMAR

Line 111 add “take” before shape

Line 296. Should “emphatic” actually be “empathic”?

Author Response

Reviewer 1

CONTENT

Nice contribution to an important topic. Your charge for adherence to greater accuracy of terminology usage is important, but is undermined by your oversight of work where this distinction occurs, such as First Sounds: Rhythm, Breath, Lullaby (Loewy et al, 2013 [ref # 58 on your ref lest]; Loewy, 2017) in its addressing of bonding and attachment in its neonate and parent/caregiver prongs respectively. One sentence of acknowledgement would strengthen the charge for more adherence in the future. 

Answer: Yes, thank you for this suggestion. We have added a sentence stressing the focus of bonding and attachment of the RBL in lines 111 ff., and cited the RBL Trainer Compendium (reference 37).

GRAMMAR

Line 111 add “take” before shape

Answer: Yes, this has been added.

Line 296. Should “emphatic” actually be “empathic”?

Answer: Yes, thank you, corrected.

Reviewer 2 Report

Reviewer Comments: Attachment and Bonding in the Neonatal Intensive Care Unit (NICU): Overlaps, differences and implications for music therapy clinical practice and research

Dear Authors,

Thank you for your work on this paper. You highlighted the importance of clear definitions for attachment and bonding which can help guide research and clinical practice. You provided a good overview of the theories from which these terms originated and provided examples the application of music therapy to the goals of attachment or bonding in NICU.

The following points can provide further clarification and strength to your paper.

  1. At the end of your introduction, include a couple of sentences clearly stating the purpose of your paper, that this paper will define attachment and bonding and a summary of why this is important. Some key statements are placed later in the paper that would strengthen your introduction and make it clearer to the reader at the beginning that the paper is not about clinical work in NICU but rather about clarifying the terms attachment and bonding (and the implications for MT). Key statements have been highlighted below.

  1. Line 35-37 is a bit awkward.

Consider rewording: How parents, both mothers and fathers, feel towards their unborn baby……

  1. Lines 52-63 you describe a number of challenges that parents may experience following a premature birth. In lines 65-58 you state that recent research indicates there is not much difference between parents of preterm birth and full term birth. Line 69 briefly mentions relationship building in NICU requires a differential view. I recommend you add a few lines here to explain this further. Did you want to highlight the contrasts in study results and the importance of assessment of parental responses to their preterm baby?   You state two very different types of data results (parents of preterm baby have a range of coping challenges versus there is no difference) and it would be helpful if you explain how this information is informing your paper by adding a couple of lines here.

  1. Line 81: I recommend you add a couple of examples of the beneficial effects of MT on immediate or short-term outcomes for infants and caregivers during their NICU stay. This would be helpful for readers who are not familiar with MT.

  1. Line 97. Recommendation: To link this more strongly to the previous paragraphs, I recommend you add a sentence such as: “Perhaps the range of study outcomes is due to the lack of clear definitions of attachment and bonding”.

  1. Attachment was defined in lines 119-123, section 3.1. However, there is no clear definition of bonding provided in lines 157-190, section 3.2.   Bonding is described, but it should also be defined.

  1. Lines 177-190. It is good that you provided a contrasting perspective.

  1. Lines 221-224. This is important. This helps to explain part of the purpose of your paper. I recommend you also mention this in your introduction at Lines 68.

  1. Lines 229-230. This is important. This helps explain part of the purpose of your paper. I recommend you also include this in your introduction section.

  1. Line 261. It is good that you mention a MT client might need a referral to the mental health care team.

  1. In attachment section: is music therapy applicable to a parent who desires attachment but the medical situation prevents them being with their child as much as they want?

  1. Line 325. This is important. It helps explain the purpose of your paper. I recommend you mention this in your introduction.

  1. In your discussion of attachment and bonding, you describe attachment first (3.1) and bonding second (3.2). In your discussion you describe bonding first (4.1) and attachment second (4.2).   I recommend you reverse them in the discussion section so that the information flow is consistent.

  1. Suggestion: you may consider a slight adjustment to your title:

Defining Attachment and Bonding: Overlaps, differences and implications for music therapy clinical practice and research in Neonatal Intensive Care Unit (NICU)

Author Response

Dear Authors,

Thank you for your work on this paper. You highlighted the importance of clear definitions for attachment and bonding which can help guide research and clinical practice. You provided a good overview of the theories from which these terms originated and provided examples the application of music therapy to the goals of attachment or bonding in NIC

The following points can provide further clarification and strength to your paper.

  1. At the end of your introduction, include a couple of sentences clearly stating the purpose of your paper, that this paper will define attachment and bonding and a summary of why this is important. Some key statements are placed later in the paper that would strengthen your introduction and make it clearer to the reader at the beginning that the paper is not about clinical work in NICU but rather about clarifying the terms attachment and bonding (and the implications for MT). Key statements have been highlighted below.

Answer: We have added a couple of sentences at the end of the introduction to make this clearer.

  1. Line 35-37 is a bit awkward.

Consider rewording: How parents, both mothers and fathers, feel towards their unborn baby……

Answer: Yes, thank you. We incorporated your suggestions and deleted the last part of the phrase.

  1. Lines 52-63 you describe a number of challenges that parents may experience following a premature birth. In lines 65-58 you state that recent research indicates there is not much difference between parents of preterm birth and full term birth. Line 69 briefly mentions relationship building in NICU requires a differential view. I recommend you add a few lines here to explain this further. Did you want to highlight the contrasts in study results and the importance of assessment of parental responses to their preterm baby?   You state two very different types of data results (parents of preterm baby have a range of coping challenges versus there is no difference) and it would be helpful if you explain how this information is informing your paper by adding a couple of lines here.

Answer: Yes, the aim was to contrast the real – but sometimes generalized – view that all parents develop mental health impairments and/or have difficulties in forming a relationship with their baby in the NICU. This is true for many parents, but not for all of them. Hence, each parent/family needs to be understood as unique. We have adapted this paragraph and added a couple of references and additional lines. Hopefully this is clearer now.

  1. Line 81: I recommend you add a couple of examples of the beneficial effects of MT on immediate or short-term outcomes for infants and caregivers during their NICU stay. This would be helpful for readers who are not familiar with MT.

Answer: Yes, we have added additional references for immediate and short-term outcomes.

  1. Line 97. Recommendation: To link this more strongly to the previous paragraphs, I recommend you add a sentence such as: “Perhaps the range of study outcomes is due to the lack of clear definitions of attachment and bonding”.

Answer: Yes, thank you. We have rephrased this sentence referring shortly to the previous paragraph.

  1. Attachment was defined in lines 119-123, section 3.1. However, there is no clear definition of bonding provided in lines 157-190, section 3.2.   Bonding is described, but it should also be defined.

Answer: Yes, we have inserted a formal definition at lines 240ff.

  1. Lines 177-190. It is good that you provided a contrasting perspective. – Great!

  1. Lines 221-224. This is important. This helps to explain part of the purpose of your paper. I recommend you also mention this in your introduction at Lines 68.

Answer: Done, thank you.

  1. Lines 229-230. This is important. This helps explain part of the purpose of your paper. I recommend you also include this in your introduction section.

Answer: Done, thank you.

  1. Line 261. It is good that you mention a MT client might need a referral to the mental health care team. – Thank you.

  1. In attachment section: is music therapy applicable to a parent who desires attachment but the medical situation prevents them being with their child as much as they want?

Answer: This is a very thoughtful question. Yes, clinically we work also with parents outside the NICU (e.g. reference 67), but to go into detail would go beyond the scope of this article. There are certainly many more specific situations to take into account, but those would probably deserve an article on its own.

  1. Line 325. This is important. It helps explain the purpose of your paper. I recommend you mention this in your introduction.

Answer: Done, thank you.

  1. In your discussion of attachment and bonding, you describe attachment first (3.1) and bonding second (3.2). In your discussion you describe bonding first (4.1) and attachment second (4.2).   I recommend you reverse them in the discussion section so that the information flow is consistent. – Thank you, excellent observation. We have now reversed the order.
  2. Suggestion: you may consider a slight adjustment to your title:

Defining Attachment and Bonding: Overlaps, differences and implications for music therapy clinical practice and research in Neonatal Intensive Care Unit (NICU)

Answer: Great suggestion, we totally agree and changed the title.

Reviewer 3 Report

The authors present an important aspect of paren child bonding in the NICU as mediated by Music Therapy. Even though I agree with the authors that there is little literature on the topic and their discussed approach from Bowlby and Ainsworth is surly suitable to match with Music Therapy, there are a couple of questions.

In order to build the argument based on the presented theories, a systematic approach thus method would be beneficial to see whether there are MT interventions besides the ones presented that deal with parent child bonding. In the current form the reader simply has not the overview needed to acknowledge the whole field and therefore the  problem at hand (there are more than the mentioned Multi-centre study, currently running to investigate parent-child bonding and multiple therapeutic methods).  

Furthermore, the authors could elaborate more on the proposed therapies, in order to make the transfer from theory to current clinical practise. This the authors do, however reading the manuscript, the reader is left with the feeling of 'wanting more' information and a more elaborate strain of thought.

Overall a valuable and informative manuscript, shedding light on an under-researched area in MT application in the NICU.

Author Response

The authors present an important aspect of paren child bonding in the NICU as mediated by Music Therapy. Even though I agree with the authors that there is little literature on the topic and their discussed approach from Bowlby and Ainsworth is surly suitable to match with Music Therapy, there are a couple of questions.

In order to build the argument based on the presented theories, a systematic approach thus method would be beneficial to see whether there are MT interventions besides the ones presented that deal with parent child bonding. In the current form the reader simply has not the overview needed to acknowledge the whole field and therefore the  problem at hand (there are more than the mentioned Multi-centre study, currently running to investigate parent-child bonding and multiple therapeutic methods).  

Answer: Yes, thank you for this comment, we totally agree. This perspective article focuses on a very specific topic within the quite large field of neonatal MT. However, since this article will be part of a special issue regarding auditory experiences, music and sound across the perinatal spectrum, the readers will have plenty of opportunities to read more on MT in the NICU. For this reason, we kept this section short, since an overview of interventions will be presented elsewhere. However, we rephrased the end of the section on “early parenting interventions in the NICU” to state that the field is much bigger and the mentioned international RCT is just one example of studies that currently research on this topic.

Furthermore, the authors could elaborate more on the proposed therapies, in order to make the transfer from theory to current clinical practise. This the authors do, however reading the manuscript, the reader is left with the feeling of 'wanting more' information and a more elaborate strain of thought.

Answer:  Thank you for this suggestion, we have reframed a couple of phrases along the document and in the transition to the clinical practice part, so we hope the manuscript reads better now. We are aware that the clinical description are not that detailed, however due to the word limits of the article we are not able to fully expand on the clinical experiences, rather they should serve as an example for how music therapists might work either on bonding and attachment. Hopefully, this motivates the reader to ‘want more’ and continue reading the other articles of this special issue!

Overall a valuable and informative manuscript, shedding light on an under-researched area in MT application in the NICU. - Thank you!